# Transcriptomic Analysis of TDP1-Knockout HEK293A Cells Treated with the TDP1 Inhibitor (Usnic Acid Derivative)

**DOI:** 10.3390/ijms26199291

**Published:** 2025-09-23

**Authors:** Alexandra L. Zakharenko, Nadezhda S. Dyrkheeva, Andrey V. Markov, Maxim A. Kleshchev, Elena I. Ryabchikova, Anastasia A. Malakhova, Konstantin E. Orishchenko, Larisa S. Okorokova, Dmitriy N. Shtokalo, Sergey P. Medvedev, Suren M. Zakian, Alexey A. Tupikin, Marsel R. Kabilov, Olga A. Luzina, Sergey M. Deyev, Olga I. Lavrik

**Affiliations:** 1Institute of Chemical Biology and Fundamental Medicine, Siberian Branch of the Russian Academy of Sciences, 8 Lavrentyeva Ave., 630090 Novosibirsk, Russia; a.zakharenko73@gmail.com (A.L.Z.); dyrkheeva.n.s@gmail.com (N.S.D.); andmrkv@gmail.com (A.V.M.); max82cll@ngs.ru (M.A.K.); lenryab@niboch.nsc.ru (E.I.R.); amal@bionet.nsc.ru (A.A.M.); medvedev@bionet.nsc.ru (S.P.M.); zakian@bionet.nsc.ru (S.M.Z.); alenare@niboch.nsc.ru (A.A.T.); kabilov@niboch.nsc.ru (M.R.K.); 2Federal Research Center Institute of Cytology and Genetics, Siberian Branch of the Russian Academy of Sciences, 10 Lavrentyeva Ave., 630090 Novosibirsk, Russia; 3AcademGene LLC, 6 Lavrentyeva Ave., 630090 Novosibirsk, Russia; larisaok123@gmail.com (L.S.O.); dmitry@novel-soft.com (D.N.S.); 4A.P. Ershov Institute of Informatics Systems, Siberian Branch of the Russian Academy of Sciences, 6 Lavrentyeva Ave., 630090 Novosibirsk, Russia; 5N. N. Vorozhtsov Novosibirsk Institute of Organic Chemistry, Siberian Branch of the Russian Academy of Sciences, 9 Akademika Lavrentieva Ave., 630090 Novosibirsk, Russia; luzina@nioch.nsc.ru; 6Shemyakin-Ovchinnikov Institute of Bioorganic Chemistry, Russian Academy of Sciences, 16/10 Miklukho-Maklaya Str., 117997 Moscow, Russia; deyev@ibch.ru; 7Department of Physical and Chemical Biology and Biotechnology, Altai State University, Pr. Lenina 61, 656049 Barnaul, Russia

**Keywords:** tyrosyl-DNA phosphodiesterase 1, topoisomerase 1, HEK293A, transcriptome, TDP1 knockout, usnic acid derivative, OL9-119, protein folding, electron transport, cell motility

## Abstract

Tyrosyl-DNA phosphodiesterase 1 (TDP1) is a key enzyme for the repair of stalled topoisomerase 1 (TOP1)-DNA complexes. Previously, we obtained HEK293A cells with homozygous knockout of the *TDP1* gene by the CRISPR/Cas9 method and used them as a cell model to study the mechanisms of anticancer therapy and to investigate the effect of *TDP1* gene knockout on gene expression changes in the human HEK293A cell line by transcriptome analysis. In this study, we investigated the effect of a TDP1 inhibitor ((R,E)-2-acetyl-6-(2-(2-(4-bromobenzyliden) hydrazinyl) thiazol-4-yl)-3,7,9-trihydroxy-8,9b-dimethyldibenzo[b,d] furan-1(9bH)-one, OL9-119, an usnic acid derivative), capable of potentiating the antitumor effect of topotecan, as well as its combination with topotecan, on the transcriptome of wild-type and TDP1 knockout HEK293A cells. OL9-119 was found to be able to reduce cell motility by decreasing the expression of a number of genes, which may explain the antimetastatic effect of this compound. Differentially expressed genes (DEGs) related to electron transport, mitochondrial function, and protein folding were also identified under TDP1 inhibitor treatment.

## 1. Introduction

Tyrosyl-DNA phosphodiesterase 1 (TDP1) is a eukaryotic DNA repair enzyme encoded by the *TDP1* gene [1] and involved in the repair of 3′-terminal DNA adducts, including stalled topoisomerase 1 (TOP1)-DNA complexes [2,3]. TOP1 is an enzyme that regulates DNA topology during repair, transcription, and other processes. To do this, TOP1 cuts one of the DNA strands, relaxes the DNA duplex, and ligates the cut strand, restoring DNA integrity [4]. During the act of catalysis, TOP1 forms a short-lived transient covalent complex (TOP1cc) between the 3′-end of DNA and tyrosine-723 (in human TOP1), which can be stabilized under certain conditions [4]. Such conditions may include DNA damage near the cross-linking site or the use of TOP1 poisons such as camptothecin and its derivatives used clinically (topotecan or irinotecan) [5]. TDP1 catalyzes the hydrolysis of the phosphodiester bond between the tyrosine residue and the 3’-DNA phosphate in TOP1cc [2]. In addition to phosphotyrosyl bonds, TDP1 hydrolyzes other blocking 3′-terminal adducts such as 3′-phosphoglycolates or antiviral or antitumor nucleosides-terminators of DNA synthesis [6]—and is also capable of hydrolyzing AP sites [7] and transphosphooligonucleotidation [8]. TDP1 is also involved in non-homologous end joining [9] and is important in oxidative phosphorylation and acts not only in the nucleoplasm but also in mitochondria [10].

An important partner of TDP1 is poly(ADP-ribose) polymerase 1 (PARP1), since PARylation recruits TDP1 to the site of DNA damage [11]. In addition to PARylation, TDP1 was shown to be phosphorylated by DNA-PK and ATM and SUMOylated in response to transcription-associated TOP1ccs in neurons [12,13,14]. Different authors highlight the significance of the role of TDP1 in neurodegeneration and also in cancer development due to the accumulation of DNA breaks during anticancer chemotherapy and radiotherapy [10,15,16,17,18].

The hypothesis that TDP1 inhibition can enhance the therapeutic effect of camptothecin derivatives and other DNA-damaging agents was put forward by the discoverers of this enzyme [2]. The hypothesis was supported by a number of studies: TDP1 deficiency in mice, TDP1 knockout in cell lines, and SCAN1 (Spinocerebellar Ataxia with Axonal Neuropathy Type 1) mutation (which reduces the activity of this enzyme) lead to hypersensitivity to camptothecin or its derivatives [19,20,21,22]. In addition, suppression of TDP1 expression by minocycline enhances the antimetastatic effect of irinotecan and increases the lifespan of experimental animals [23]. Conversely, in cells with increased TDP1 expression, camptothecin causes less DNA damage [24,25]. Moreover, in intestinal tumors overexpressing TDP1, the response to irinotecan therapy is less [19]. It was shown that the inhibition of TDP1 can restore sensitivity to topotecan [26]. These data suggest that the combination of anticancer drugs and TDP1 inhibitors may significantly improve the efficacy of chemotherapy.

In our works we have demonstrated that the TDP1 inhibitor added in combination with topotecan reveals a synergistic effect both in vitro and in vivo [3,17,27]. These compounds enhance both the antitumor and antimetastatic effects of topotecan, which is of particular interest due to the importance of the problem of metastasis [27,28,29]. Thus, TDP1 is regarded as a potential therapeutic target in cancer therapy.

Previously, we examined the effect of TDP1 knockout on the transcriptome of HEK293A cells and found that TDP1 is important in different processes such as cellular contact communication, spermatogenesis, protein synthesis and degradation, and mitochondrial function [30]. In the present study, we compared differentially expressed genes (DEGs) induced by treatment with a TDP1 inhibitor, OL9-119 ((R,E)-2-acetyl-6-(2-(2-(4-bromobenzyliden)hydrazinyl)thiazol-4-yl)-3,7,9-trihydroxy-8,9b-dimethyldibenzo[b,d] furan-1(9bH)-one) [31], topotecan, or their combination in wild-type and TDP1 knockout cells.

The aim of this work was to investigate the effect of the TDP1 inhibitor OL9-119, topotecan, and their combination on the transcriptome of HEK293A cells and to clarify the role of TDP1 in the response to these compounds. We found that the TDP1 inhibitor, an usnic acid (UA) derivative OL9-119, was able to reduce cell motility by decreasing the expression of a number of genes, which may explain the antimetastatic effect of this compound. The number of DEGs upon treatment with OL9-119 coincides with the DEGs upon TDP1 knockout. DEGs associated with electron transport and mitochondrial function, as well as protein folding, were also identified.

## 2. Results

We have previously shown that compound OL9-119 is able to enhance the cytotoxic/antiproliferative effect of topotecan in vitro and the antitumor and antimetastatic effect in vivo [31]. Here, we aim to shed light on the mechanism of action of this compound and look for its potential side effects by analyzing the changes in transciptome induced by OL9-119.

TDP1 knockout sample preparation and characterization were described earlier, as well as the effect of TDP1 knockout on the transcriptome of HEK293A cells [30]. The effect of 0.1% DMSO as an OL9-119 solvent on gene transcription in HEK293A cells was described in [32]. It should be noted that the effect was insignificant, which is inconsistent with the data of other researchers [33,34,35].

We analyzed the effect of TDP1 inhibitor OL9-119, antitumor drug topotecan, and their combination by bulk transcriptome sequencing of the control HEK293A WT cell line and three TDP1-KO biological replicates: C6, G6, and F7 cell clones, in four technical repeats. As an inhibitor, we used a UA derivative, OL9-119, which is capable of sensitizing tumor cells in vitro and in vivo to the action of topotecan (compound 20d in the article [31] and compound 4 in the article [27]).

Principal component analysis (PCA) was used to determine the data clustering, and it was found that three TDP1-KO cell clones (C6, G6, F7) formed clearly defined clusters on the PCA plot (Figure 1). Also, within each cluster, the types of treatment (TDP1 inhibitor, topotecan, or their combination) were clearly separated. For comparison, we also took transcriptome data from PARP1 knockout HEK293A cells [32]. On the PCA plot, the PARP1-KO samples are very different from the others. The different response patterns for the resulting clones TDP1-KO clones C6, G6, and F7 may be due to epigenetic differences, different degrees of compensation for TDP1 deficiency, and variations in the integration of the CRISPR construct.

### 2.1. The Effect of OL9-119 on the Transcriptome of HEK293A Cells

To understand the molecular mechanism of action of OL9-119, its effect on the transcriptome profile of HEK293A cells was investigated using RNA-seq. To examine genes under the OL9-119 effect that changed their expression on the transcription level, we identified genes with adjusted *p*-values < 0.05 and |fold change| > 1.5 (log2FoldChange > 0.59, Appendix A). As shown in Figure 2A, OL9-119 at 5 µM induced changes in the expression of 1193 genes, of which 685 and 508 were up- and down-regulated, respectively. The top 5 up-regulated differentially expressed genes (DEGs) included *RB1-DT*, a long non-coding RNA that controls cell proliferation [36] *ADM2*, a sensor of mitochondrial respiratory chain blockade [37] *H2AC8*, which encodes a core nucleosome component HIST1H2AE [38]; *TEX22*, which is associated with insulin signaling [39]; and the lncRNA *METTL14-DT* of unknown function.

In the case of down-regulated DEGs, the most pronounced suppression was found for regulators of cell homeostasis, including *CACNA1F*, encoding a voltage-dependent calcium channel involved in the regulation of cell motility and cell division [40]; *KRT34*, a known sensor of F-actin depolymerization [41]; *PRAMEF9*, a component of the Cul2-RING ubiquitin ligase complex that degrades aberrant proteins [42]; the helicase *DDX43* associated with chromatin remodeling [43]; and the glycosyltransferase *MGAT4C* that prevents the degradation of CD133, a known cancer stem cell marker [44] (Figure 2A).

Further functional annotation of the DEGs revealed a tight cluster of oxidative phosphorylation-related terms, enriched predominantly by down-regulated genes, clearly demonstrating the mitochondria-targeting effect of OL9-119 (Figure 2B). This effect was also accompanied by suppression of genes involved in regulation of splicing and protein folding; and activation of stress-related processes such as EIF2AK1 signaling, membrane amino acid transport, and transcriptional response (Figure 2B). These results indicate that HEK293A cells attempt to adapt to the effects of OL9-119 and that mitochondrial dysfunction appears to be one of the major consequences of exposure to this compound. Indeed, further analysis of OL9-119-sensitive DEGs using the Connectivity Map approach independently confirmed this hypothesis by identifying a significant similarity of the OL9-119 transcriptional response to that of carbonyl cyanide 3-chlorophenylhydrazone (CCCP), a known disruptor of mitochondrial membrane potential [45] (Figure 2C). Furthermore, gene set enrichment analysis (GSEA) also revealed a close association of down-regulated genes in the OL9-119-treated group with electron transport chain and oxidative phosphorylation (Figure 2D), confirming the mitochondria-targeting effect of OL9-119. In addition, Connectivity Map analysis demonstrated similarities between the transcriptomic profile changes induced by OL9-119 and those induced by dequalinum, azacytidine, nutlin-3, and cyclosporin A, which inhibit protein kinase C, DNA methyltransferase, MDM2, and calcineurin, respectively (Figure 2C). This finding demonstrates a probable multi-targeting effect of OL9-119, as expected due to its natural origin.

#### 2.1.1. Topological Analysis of OL9-119-Sensitive Regulome

To further analyze the transcriptomic response of HEK293A cells to OL9-119 in more detail, the gene association network was reconstructed from the identified DEGs using the STRING database, and its topological analysis was performed. As shown in Figure 3A, OL9-119-sensitive genes formed a dense network consisting of 454 nodes and 1187 edges, with down-regulated *HSP90AA1* and *ACTB*, encoding heat shock protein Hsp90 and actin beta, and up-regulated *JUN* and *EGFR*, encoding transcription factor c-Jun and epidermal growth factor receptor EGFR, respectively, occupying the most central (hub) positions within the network. The list of DEGs forming the OL9-119-sensitive regulome, the degree of their change, and the number of connections in the network are given in Appendix A.

Interestingly, the majority of the top 7 most interconnected DEGs were regulators of the mitochondrial respiratory chain, including *NDUFAB1, COX5A, UQCRQ*, and *ATP5MF*, which is consistent with the mitochondria-targeting effect of OL9-119 revealed above in the functional (Figure 2B), Connectivity Map (Figure 2C), and gene set enrichment (Figure 2D) analysis. In the case of up-regulated DEGs, the hub positions were also revealed for *H4C14* and *H2AC6* encoding histone proteins, anti-apoptotic BCL2, and stress response sensors NFKBIA and ATF3 (Figure 3B).

Furthermore, using the molecular complex detection (MCODE) algorithm, three highly interconnected gene modules were identified within the OL9-119 susceptible regulome (Figure 3C), whose functional analysis revealed their association with the electron transport chain, mRNA splicing, and protein refolding (Figure 3D). Thus, the high consistency of these results with the findings described above confirms the negative effect of OL9-119 on mitochondrial homeostasis and the sensitivity of the splicing and protein chaperone system to this compound.

#### 2.1.2. The Relation of TDP1-Inhibiting Activity of OL9-119 with OL9-119-Susceptible Hub Genes

As it was shown previously, OL9-119 is a potent inhibitor of TDP1, directly blocking its activity at low micromolar concentrations (IC_50_ = 0.03 µM) [31]. Although the performed analysis of DEGs did not reveal any terms associated with DNA repair, we hypothesized that the observed change in transcriptome profile in HEK293A cells under OL9-119 treatment (Figure 3A) might be a response to its direct interaction with TDP1. To evaluate this issue, the complex protein-gene network consisting of TDP1, its first protein neighbors from the STRING database, and OL9-119-sensitive DEGs forming edges with them was reconstructed (Figure 4). Almost all proteins that STRING identified as TDP1 partners (yellow circles in Figure 4) are involved in the repair of single-strand DNA breaks [46]. In particular, we have previously shown a direct interaction of TDP1 with such participants in base excision repair as APE1, PARP1, Polβ, and XRCC1 [47]. As expected, the greatest number of connections with OL9-119-dependent DEGs was found in such an important cellular regulator as PARP1. On the one hand, PARP1 controls TDP1-dependent repair of TOP1cc adducts [48] on the other hand, it regulates a number of important cellular processes, such as cell death (*JUN, BCL2, SUMO1*), proliferation (*EGFR*), signaling (*ACTB, ABL1*), and differentiation (*ABL1*) [49].

As expected, a number of hub genes, including down-regulated *ACTB* and up-regulated *JUN*, *EGFR*, *H4C14*, and *BCL2*, although not directly linked to TDP1, formed edges with its partners with a high confidence score, confirming that the observed transcriptional response of OL9-119-treated HEK293A cells may be TDP1-dependent (Figure 4). Given the absence of oxidative phosphorylation-related hub genes within this network (Figure 4), we suggest that the mitochondria-targeting effect of OL9-119 may be a TDP1-independent response that requires further detailed studies.

#### 2.1.3. OL9-119 Effects on Transcriptome in the Context of TDP1 Knockout

Next, to estimate which of the above-described processes in OL9-119-treated cells are off-target effects of the compound, probably determined by the natural UA scaffold, two independent approaches were applied.

First, we overlaid DEGs determined in OL9-119-treated wild-type cells with DEGs specific to each TDP1 knockout cell strain (C6, F7, and G6; knockout versus wild-type cells). As shown in the Venn diagram (Figure 5A), the effect of OL9-119 on the HEK293A cell transcriptome is quite similar to the TDP1 knockout effect. Only 34% (411 genes) of all OL9-119-susceptible DEGs are unique to the compound under study. Considering that these genes may determine the off-target effects of OL9-119, we performed a functional annotation of these genes. As shown in Figure 5B, the unique DEGs not associated with the TDP1-targeting effect of OL9-119 were related to RNA processing and protein folding, as mentioned earlier in Section 2.1 (Figure 2B).Second, to distinguish more precisely between TDP1-independent and TDP1-dependent processes modulated by OL9-119, RNA sequencing was performed on G6, F7, and G6 knockout cells treated with the tested compound (a table of DEGs for all cell types is provided in Appendix A). Further GSEA of obtained DEGs (OL9-119-treated vs. untreated cells) revealed 17 processes common for both knockout clones and wild-type cells, which indicates that OL9-119 modulates them independently of TDP1 status (off-target effect) (Figure 5C). Among these processes, three functional terms—namely, oxidative phosphorylation, protein processing in the endoplasmic reticulum (folding), and the spliceosome—have been previously identified (Figure 2B,D, Figure 3B, and Figure 5B). In turn, TDP1-dependent processes, which OL9-119 modulated in only wild-type cells and which were absent in knockout clones, involved various metabolic processes, cell proliferation, inflammation-related terms, and cell differentiation (Figure 5D).

Thus, bioinformatic analysis revealed that, although HEK293A cells exhibit a similar transcriptional response to OL9-119 and TDP1 knockouts (Figure 5A), the ability of the tested compound to suppress gene expression related to oxidative phosphorylation, RNA processing, and protein folding appears to be TDP1-independent and associated with the UA backbone. The effect of UA on living cells has been well studied. In particular, such effects include a decrease in mitochondrial membrane potential and uncoupling of oxidative phosphorylation, ER stress, and glutathione and ATP depletion [50].

### 2.2. Effect of OL9-119 on Cell Motility

Cell migration plays a crucial role in cancer invasion and metastasis [51]. We previously noted that combined treatment of mice with Lewis’s carcinoma (topotecan + OL9-119) led to a decrease in the number of lung metastases by almost an order of magnitude [31]. The mechanism of the antimetastatic action of OL9-119 is unknown. For another UA derivative, OL9-116, we have shown that this compound in combination with topotecan acted directly on metastatic cells in the lungs, rather than on the production of metastases by the primary node [28]. In the previous paper, we found that knockout of TDP1 resulted in decreased cell motility, which was reflected in slower healing of the scratches and changes in the expression of the *CTGF (CCN2)–THBS1–COL6A3* genes responsible for cell adhesion [30]. That is, metastasis can be affected by TDP1 deficiency, either by knockout or inhibition. Therefore, it was interesting to examine the effect of OL9-119 on prometastatic gene expression at the in vitro level.

In the present study, we compared DEGs in all cell types treated with OL9-119 and in TDP1 knockout cells and found a set of metastasis-associated genes whose expression was altered in response to TDP1 deficiency: in knockout cells without inhibitor treatment and in wild-type cells treated with TDP1 inhibitor (Table 1). The genes *MTSS1* (adhesion, negative regulation of the cell cycle), *MMP9* (matrix metalloproteinase, except C6) are up-regulated; *RPSA* (laminin receptor 1, extracellular matrix glycoprotein), *MMP2* (except C6), *NME4* (Nucleoside Diphosphate Kinase 4, except G6), and *MDM2* (negative regulation of cell proliferation; did not change in WT cells) are down-regulated. The expression of the *VEGFA* gene (regulation of the cell cycle, positive regulation of cell proliferation, growth factor) increased both under the effect of TDP1 knockout and under the effect of the TDP1 inhibitor. Moreover, it changed most strongly in wild-type cells under the effect of OL9-119; under the effect of TDP1 knockout (compared with the wild type) and in knockout cells under the effect of OL9-119 (compared with TDP1-/- cells without treatment), it also changed. It looks as if the effect of knockout and inhibitor was additive.

There are also three genes whose expression changed only in knockout cells, but not under the action of the TDP1 inhibitor: growth factors *TGFB1* and *HGF* and a protein required for epithelial–mesenchymal transition *MDM2*. It can be assumed that the presence of the TDP1 protein, and not its activity, is required for their expression to change.

Because OL9-119 had an ambiguous effect on the expression of metastasis-related genes, we examined its effect on HEK293A WT cell motility. For this, we determined the velocity of HEK293A WT cell migration on cultural plastics. Cell motility was assessed for 20–25 randomly selected cells (one in each field). As a result, we evaluated the mean velocity of WT cells treated with OL9-119. The results are shown in Figure 6. The migration rate of cells treated with OL9-119 statistically significantly differs from those treated with DMSO, and a tendency towards further slowing of migration is observed with increasing inhibitor concentration.

The parent compound UA is known to inhibit cell migration through DNA damage response activation and modulation of protein signaling pathways [52,53,54,55,56] apparently, its derivative OL9-119 also has this property, which makes it a valuable component of antitumor therapy. The mechanisms of anti-migration action of OL9-119 need to be studied.

It should be noted that in vivo OL9-119 in monotherapy has neither antitumor nor antimetastatic effect [31]. It is likely that the administration scheme used (four times intragastrically at a single dose of 100 mg/kg) is not optimal for the manifestation of the intrinsic effect of OL9-119.

### 2.3. The Effect of OL9-119 on Mitochondrial Membrane Potential

As noted above, the effect of OL9-119 on the transcriptome of HEK293A cells was accompanied by the downregulation of genes associated with mitochondrial function, which was independently identified using functional annotation (Figure 2B), Connectivity Map analysis (Figure 2C), GSEA (Figure 2D), and clustering of the OL9-119 regulome (Figure 3C,D). To experimentally verify the mitochondria-targeting effect of OL9-119, we evaluated its effect on mitochondrial membrane potential (ΔψM) in HEK294A cells using a fluorescent potentiometric JC1 probe. This dye accumulates in polarized mitochondria in the form of red fluorescent aggregates, but when ΔψM decreases, it leaves the mitochondria and enters the cytoplasm, which is accompanied by a shift in the fluorescence spectrum into the green region. As shown in Figure 7A, incubation of cells with OL9-119 at 5 µM (IC_50_) for 24 h led to significant changes in the cell population (Figure 7A), increasing the green-to-red fluorescence ratio by 3.3 times compared to the control (Figure 7B). These data indicate that OL9-119-induced dissipation of ΔψM, confirming the ability of OL9-119 to disrupt mitochondrial function and, therefore, the plausibility of our bioinformatics results.

### 2.4. Ultrastructural Changes in HEK293A Cells Under the Effects of Topotecan and OL9-119

Since we identified gene modules related to the electron transport chain and protein synthesis by studying the OL9-119 susceptible regulome (Figure 3C,D), we investigated how OL9-119 could affect cell ultrastructure. We also studied the effect of topotecan and the combination of the two drugs. Incubation of HEK293A cells with topotecan or OL9-119, or with a combination of the two, resulted in pronounced ultrastructural changes after 24 h, compared to intact cells (Figure 8A,D).

The most striking change in cells incubated with topotecan was loss of characteristic mitochondrial structure: mitochondria swelled, cristae collapsed, and matrix formed clusters of structureless material (Figure 8B,E,F). Although we detected DEGs among genes associated with oxidative phosphorylation under the influence of OL9-119 (Figure 2B and Figure 3D), and we observed the mitochondria-targeting effect of compound OL9-119 after 24 h treatment experimentally (Figure 7), we found evidence of such stress in ultramicrographs only under the influence of topotecan.

As shown earlier, intake of 200 mg of UA per day for 14 days by B6C3F1 mice increased the expression of 37% of genes associated with mitochondrial oxidative phosphorylation [57]. After treatment of OL9-119 cells, we also observed significant changes in the expression of these genes, but it is predominantly reduced. In addition, in the case of HEK293A cells and the UA derivative OL9-119, unlike the compensatory changes in the expression of genes responsible for fatty acid/lipid metabolism and the Krebs cycle observed with UA in mice [58].

In addition to influencing the expression of genes of the mitochondrial respiratory chain (MRC), UA also induced ROS production via inhibition of complex I and complex III of the MRC and via reducing Nrf2 stability through disruption of the PI3K/Akt pathway [54]. In the same work it was shown that UA causes apoptosis of lung squamous cell carcinoma (LUSC) cells [59], as did OL9-119 in our work [60], although OL9-119 did not significantly alter the expression of apoptosis-associated genes. It is possible that the mechanism of apoptosis caused by OL9-119, as well as UA, is associated with disrupting the MRC, signs of which we observe in the ultramicroscopic images of cells in Figure 8B,E,F as a change in the structure of the mitochondria.

During incubation with topotecan, a pronounced expansion of rough endoplasmic reticulum (ER) cisternae was observed in the cytoplasm, indicating the development of ER stress (Figure 8C,G). Topotecan has previously been shown to increase the expression of ER stress-related genes such as *ASNS*, *ATF3*, *CARS*, *CDKN1A*, *CEBPG*, *CHAC1*, *CTH*, *DDIT3*, *RNF19B*, *TRIB3*, and *WDR45* in the NCI-60 lines [59]. Three of these genes, namely *ATF3*, *CEBPG*, and *CHAC1*, were also upregulated in our experiment, with *ATF3* almost 4-fold and *CHAC1* more than 5-fold (Appendix A).

The destructive changes observed in cells incubated with a mixture of OL9-119 and topotecan were a combination of the damage caused by each agent separately (Figure 8H,I). In addition to these changes, there was an increase in the number of late endosomes and lysosomes, as well as electron-dense areas appearing on the organoid structure’s membrane, which is indicative of peroxidation processes (Figure 8E,G–I).

The results obtained showed that the development of destructive changes in HEK293A cells upon exposure to topotecan and OL9-119 occurred via different pathways. In the event of joint exposure, these disorders combine to eventually lead to more pronounced destructive changes. Thus, at the ultrastructural level, the synergistic effect of the OL9-119 and topotecan compounds is evident.

## 3. Materials and Methods

### 3.1. Total RNA Preparation and Transcriptome Sequencing

TDP1-KO cells were obtained as described previously [30]. HEK293A WT and TDP1-KO cells were grown 1–2 million per well in a six-well plate for 20–21 h to double the number of cells in the well, four replicates for each sample. The RNA samples for the gene expression studies were prepared for four cell lines: (1) WT and three TDP1-KO cell clones (2) C6; (3) G6; and (4) F7 with different treatments: (1) no treatment; (2) 0.1% DMSO; (3) 100 nM (for WT HEK293A) or 50 nM (for PARP1-KO HEK293A) Tpc + 0.1% DMSO; (4) 5 µM OL9-119 dissolved in 0.1% DMSO; (5) 100 nM (for WT HEK293A) or 50 nM (for PARP1-KO HEK293A) Tpc + 5 µM OL9-119. Total RNA was isolated from cells by Trizol and PureLink RNA Mini Kit (Invitrogen, Carlsbad, CA, USA) and on-column DNase I Digestion Set (Sigma-Aldrich, Saint Louis, MO, USA). Isolated RNA was measured on Nanodrop 1000 spectrophotometer (Thermo Scientific, Waltham, MA, USA), and the quality was estimated by means of measuring RNA Integrity Number (RIN) on Agilent Bioanalyzer 2100 with Agilent RNA 6000 Pico Kit (Agilent, Santa Clara, CA, USA).

To isolate poly(A)+ RNA from total RNA, the NEBNext Poly(A) mRNA Magnetic Isolation Module (NEB, Ipswish, MA, USA) was used in the SB RAS Genomics Core Facility (ICBFM SB RAS, Novosibirsk, Russia). The quality of enriched poly(A) + RNA was assessed on an Agilent Bioanalyzer 2100 using an Agilent RNA 6000 Pico Kit (Agilent, Santa Clara, CA, USA). The library was prepared from poly(A) + RNA by the MGIEasy RNA Directional Library Prep Set (MGI Tech Co., Ltd., Shenzhen, China). DNA libraries were sequenced with 100 bp paired-end reagents with 30 million coverage per sample on MGIseq 2000 (MGI Tech Co., Ltd., Shenzhen, China).

### 3.2. Bioinformatic Analysis

The obtained reads were aligned to the human genome (hg38, ensembl v38.93) using STAR (v2.7.8) [61]. Quality control of the reads was performed using FastQC (0.11.9) [62], the “infer_experiment.py” module of RSeQC (v3.0.1) [63], and the “CollectRnaSeqMetrics” module from Picard (v2.25.2) [64]. All results were consolidated into a comprehensive report using the MultiQC package (v1.9) [65]. Read quantification was carried out using STAR with the “quantMode GeneCounts” option. Downstream analysis of the expression matrix has been performed in the DESeq2 package (v1.31.2) [66]. Subsequently, low-expression genes (sum of reads across all samples less than 10) were filtered from the resulting expression matrix. Vst normalization was applied to the expression matrix to facilitate downstream analysis, including principal component analysis (PCA). Differentially expressed genes were identified using the Wald test, considering genes with a *p*-value less than 0.01 after adjusting for multiple comparisons. The EnchancedVolcano package (v1.6) [67] and VolcaNoseR [68] were used for visualization of the results. To gain insights into the biological context of the differentially expressed genes, signaling pathway enrichment analysis was performed using the fgsea package (v1.17.0) [69] with reference to the KEGG database, as well as the ClueGO plugin (v2.5.9) in Cytoscape (v3.10.3) (in the latter case, functional annotation was performed using Gene Ontology (Biological Processes), KEGG, and WikiPathways databases). Reconstruction of gene association networks and their clusterization were performed using stringApp (v2.2.0) and MCODE (v2.0.3), respectively. GSEA was performed using the WebGestalt 2024 v1.0 tool [70].

### 3.3. Analysis of HEK293A WT Cell Motility Under OL9-119 Treatment

This assay was performed on HEK293A WT cells treated with OL9-119 at different concentrations (5 to 15 µM) by real-time cell imaging with a Cell- IQ automated cell culture and analysis system (CM Technologies, Lewisville, TX, USA). The experiments were carried out in duplicate. For the imaging, 3000 cells were seeded per well of a 24-well plate. At 36 h after the seeding, the cell motility was analyzed by culturing in the Cell-IQ system for 2 days at 37 °C; 5% humidified CO_2_ was pumped directly through the culture plate in the following regimen: 15 min pump on, 15 min pump off, and initial gas supply for 30 min. The cells were imaged over 43 h with a Nikon CFI Plan DL Fluorescence objective (×10), 3 times per hour, 10–15 fields per well of a 24-well plate, and 3 wells for each concentration of OL9-119. The cells were treated with OL9-119 immediately before being placed in the Cell- IQ. Cells in control wells were treated with vehicle (1% DMSO). Cell tracking on the time-lapse images was performed manually and analyzed using Cell-IQ Analyzer 4 Pro-Write Version AN4.3.0 software. We analyzed trajectories of cells that did not contact other cells over the period of tracking.

### 3.4. Analysis of Mitochondrial Membrane Potential

HEK293A WT cells were seeded in a 12-well plate and cultured overnight at 37 °C and 5% CO_2_. The cells were then treated with OL9-119 at X µM for 24 h. Afterward, the cells were harvested by trypsinization, resuspended in JC-1-containing PBS (5 µg/mL; 10^6^ cell/well), and incubated in a CO_2_ incubator for 30 min. The cells were then washed with PBS and analyzed by flow cytometry, using a NovoCyte Flow Cytometer (ACEA Biosciences Inc., San Diego, CA, USA). For each sample, 10,000 events were collected.

### 3.5. Transmission Electron Microscope (TEM) Examinations

Suspensions of intact HEK293A cells, as well as cells treated with topotecan and OL9-119, were fixed with 4% formaldehyde in Hank’s solution overnight. After three washes with Hank’s solution, the samples were fixed in a 1% osmium tetraoxide (EMS, Hatfield, PA, USA) in Hank’s solution for one hour, followed by three more washes with the solution. The samples were then dehydrated according to the standard protocol and embedded in an epon–araldit mixture. Ultrathin sections were then prepared from the obtained solid blocks on a Leica EM UC7 ultramicrotome (Leica, Wetzlar, Germany). The ultrathin sections were then contrasted with lead citrate and uranyl acetate solutions (both EMS, Hatfield, PA, USA) and examined using a JEM-1400 TEM (JEOL, Tokyo, Japan). Images were obtained using a Veleta digital camera (EM SIS, Muenster, Germany).

## 4. Conclusions

In this study, we aimed to investigate the effects of OL9-119, topotecan, and their combination on the transcriptome of wild-type and TDP1-null cells and, possibly, to elucidate the mechanism by which OL9-119 enhances the antitumor and antimetastatic activity of topotecan in vivo. In our previous work, we showed that TDP1 knockout causes changes in a number of cellular processes, such as cellular contacts, communication, spermatogenesis, protein synthesis and degradation, and mitochondrial function [30]. In both the previous [30] and current studies, we found no effect of TDP1 deficiency on the expression of DNA repair genes. However, reconstruction of the network consisting of OL9-119-sensitive DEGs forming edges and first protein neighbors of TDP1 from the STRING database revealed a number of hub genes that, although not directly linked to TDP1, formed edges with its partners with a high confidence score. Among them, there are a number of genes that play key roles in the regulation of cell survival, such as ACTB (cell shape, movement, and signaling); JUN and BCL-2 (cell death); EGFR (cell growth, survival, proliferation, and differentiation); and others. The greatest number of connections with OL9-119-dependent DEGs was found in such an important cellular regulator as PARP1, which controls both TOP1cc adduct repair and a number of cell processes such as proliferation, signaling, differentiation, and cell death.

The results obtained in the present study show that treatment with the TDP1 inhibitor OL9-119, like knockout, disrupts mitochondrial and rough ER function (Figure 2 and Figure 8). This is indicated by both bioinformatics analysis and experimental data. Based on the absence of oxidative phosphorylation-related hub genes within the network consisting of TDP1 partners and the OL9-119-associated regulome, the mitochondria-targeting effect of OL9-119 is most likely related to the ability of OL9-119 to uncouple oxidative phosphorylation “inherited” from UA, as well as to the effect of OL9-119 on the expression of a number of mitochondria-associated genes, rather than to inhibition of TDP1.

Treatment of OL9-119 cells resulted in a concentration-dependent decrease in migration rate (Figure 6). In addition, OL9-119 has a significant effect on the expression of metastasis-related genes, often in line with the effect of TDP1 knockout. This indicates the potential for the use of TDP1 inhibitors as antimetastatic drugs.

## Figures and Tables

**Figure 1 ijms-26-09291-f001:**
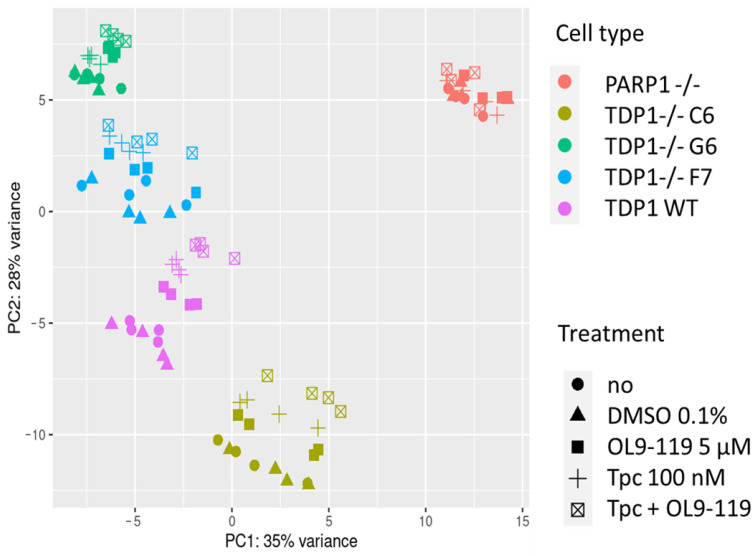
Dataset clustering by visualizing the transcriptional pattern of cells (HEK293A WT and PARP1-/- or TDP1-/- treated with DMSO, OL9-119, topotecan (Tpc), or the combination) with differences/similarities in gene expression among samples on the PCA plot.

**Figure 2 ijms-26-09291-f002:**
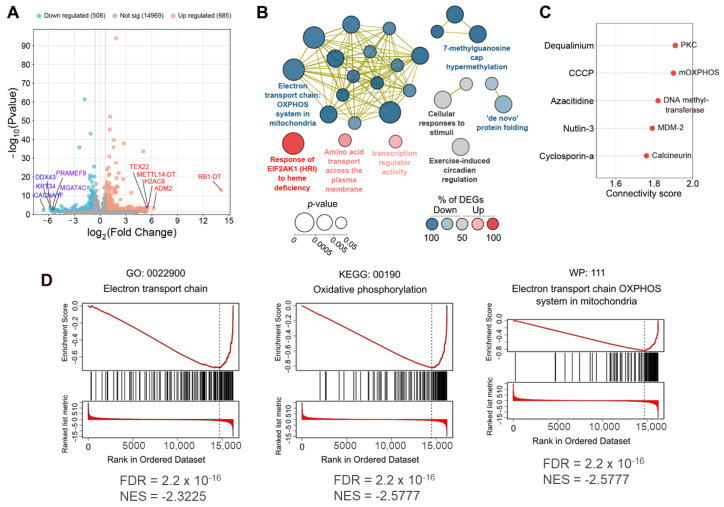
Bioinformatic analysis of whole-genome RNA sequencing of OL9-119-treated HEK293A cells. (**A**) Volcano plot showing differentially expressed genes (DEGs) in OL9-119-treated HEK293A cells compared to control cells. The top 5 up- and down-regulated DEGs characterized by the largest expression change are highlighted separately. (**B**) Functional analysis of DEGs using the ClueGO platform in the Cytoscape bioinformatics environment. (**C**) Connectivity Map analysis results of the top 100 up- and down-regulated DEGs in HEK293A cells. The connectivity score is a measure of the similarity of OL9-119 to known pharmaceutics in the context of transcriptome alteration profile (protein targets of these compounds are indicated on the right). (**D**) Gene set enrichment analysis (GSEA) of OL9-119-susceptible DEGs according to Gene Ontology (GO), Kyoto Encyclopedia of Genes and Genomes (KEGG), and WikiPathways (WP) databases. NES—normalized enrichment score.

**Figure 3 ijms-26-09291-f003:**
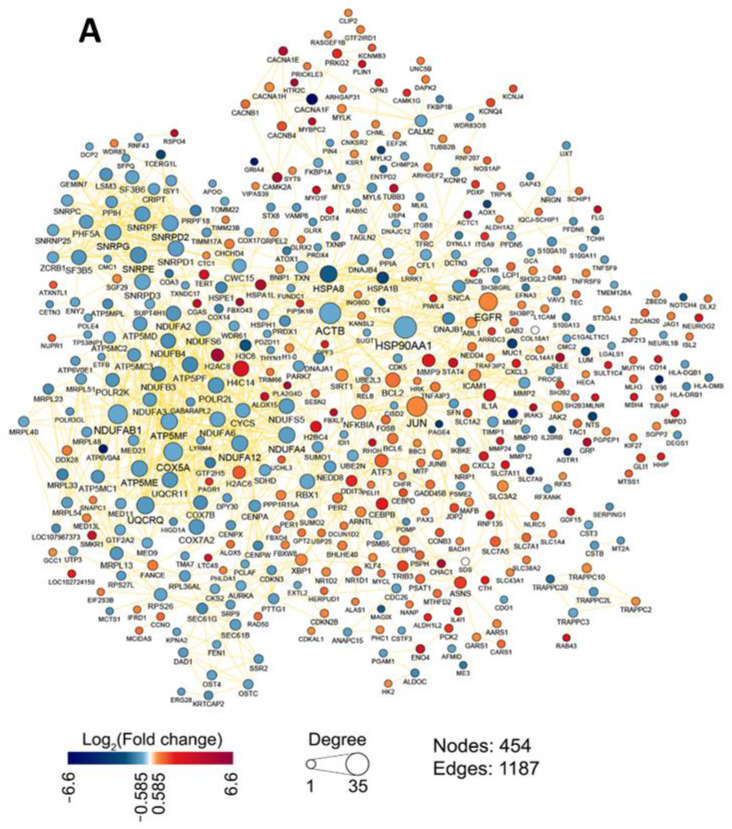
OL9-119-sensitive regulome. (**A**) The gene association network consisting of OL9-119-sensitive DEGs. The edges between DEGs were retrieved from the STRING database (confidence score > 0.7). (**B**) Topological features of the top seven down- and up-regulated hub genes. The corresponding fold change value is indicated to the right of the marker. (**C**) MCODE clustering of OL9-119 regulome. The modules with scores greater than eight were analyzed. (**D**) Functional annotation of genes from MCODE clustering by ClueGO plugin.

**Figure 4 ijms-26-09291-f004:**
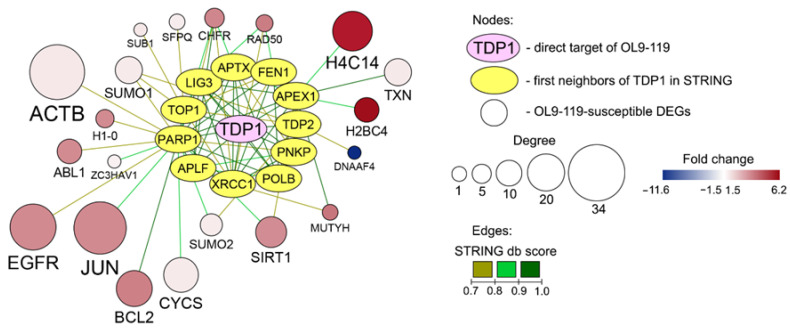
The association of TDP1 with the OL9-119-associated regulome. Yellow ovals indicate the first neighbors of TDP1 (TDP1 partner proteins) obtained from the STRING database (STRING db score > 0.7). Round nodes indicate DEGs (OL9-119 vs. control) associated with TDP1 and its partner proteins. Degree means the connectivity level, the number of connections with neighbors in the network.

**Figure 5 ijms-26-09291-f005:**
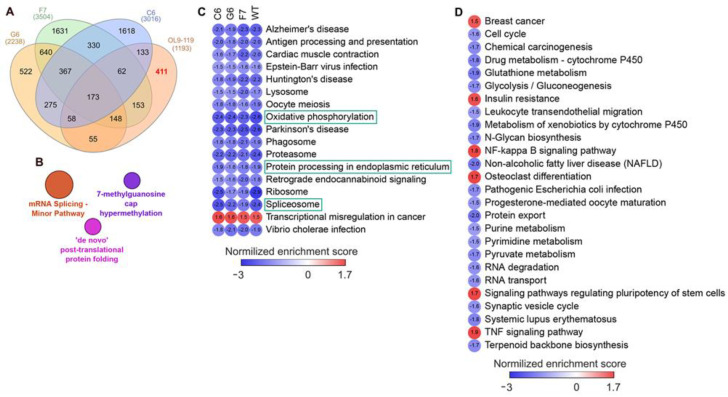
Comparison of the effect of OL9-119 cells on the HEK293A transcriptome with the effect of TDP1 knockout. (**A**) Venn diagram showing the similarity in transcriptional response between HEK293A cells with TDP1 knockouts and HEK293A WT cells treated with OL9-119. The number of DEGs that determine the TDP1-independent off-target effect of OL9-119 is marked in red. (**B**) Functional analysis of the 411 OL9-119-susceptible DEGs determining the off-target effect of OL9-119. The analysis was performed using the ClueGO plugin in Cytoscape. (**C**) GSEA of DEGs in wild-type and TDP1 knockout HEK293A cells (OL9-119-treated vs. 0.1% DMSO-treated cells) according to the KEGG database. Only the statistically significant functional terms common to wild-type cells and all knockout strains are shown. The key OL9-119-susceptible processes mentioned above are circled in green. (**D**) Processes modulated by OL9-119 in wild-type HEK293A cells only (TDP1-dependent processes). GSEA of DEGs was performed using the fgsea package in R (KEGG database).

**Figure 6 ijms-26-09291-f006:**
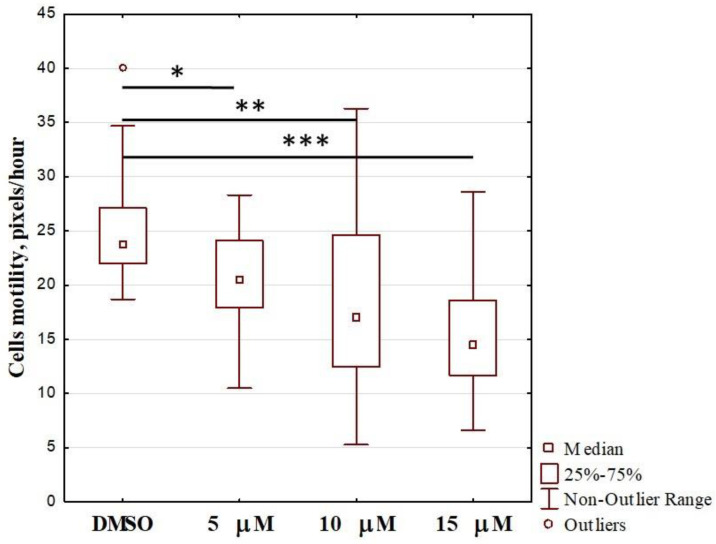
Effect of OL9-119 on the motility of HEK293A cells. The X-axis shows the concentrations of OL9-119 in the cell culture medium. Control cells were incubated in 0.1% DMSO. *p* values: * −0.0034; ** −0.000018; *** −0.000008 according to Tukey’s test.

**Figure 7 ijms-26-09291-f007:**
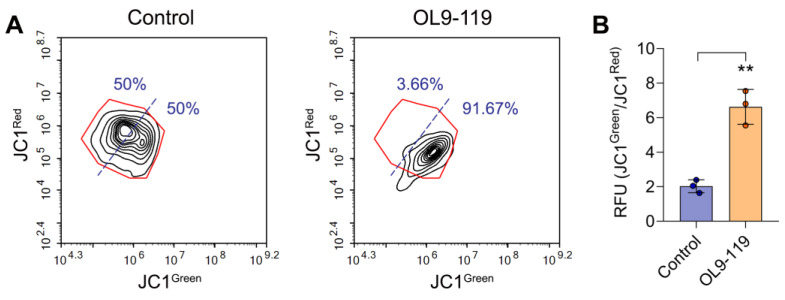
Effect of OL9-119 on mitochondrial membrane potential in HEK293A cells. (**A**) Cytograms showing the distribution of cells based on red and green fluorescence intensity. Red fluorescence corresponds to JC1 aggregates, which are specific to mitochondria, while green fluorescence corresponds to JC1 monomers, which are specific to the cytoplasm. (**B**) A bar plot showing an increase in the proportion of green-to-red JC1 fluorescence under OL9-119 treatment, indicating dissipation of mitochondrial membrane potential. ** *p*-value < 0.01.

**Figure 8 ijms-26-09291-f008:**
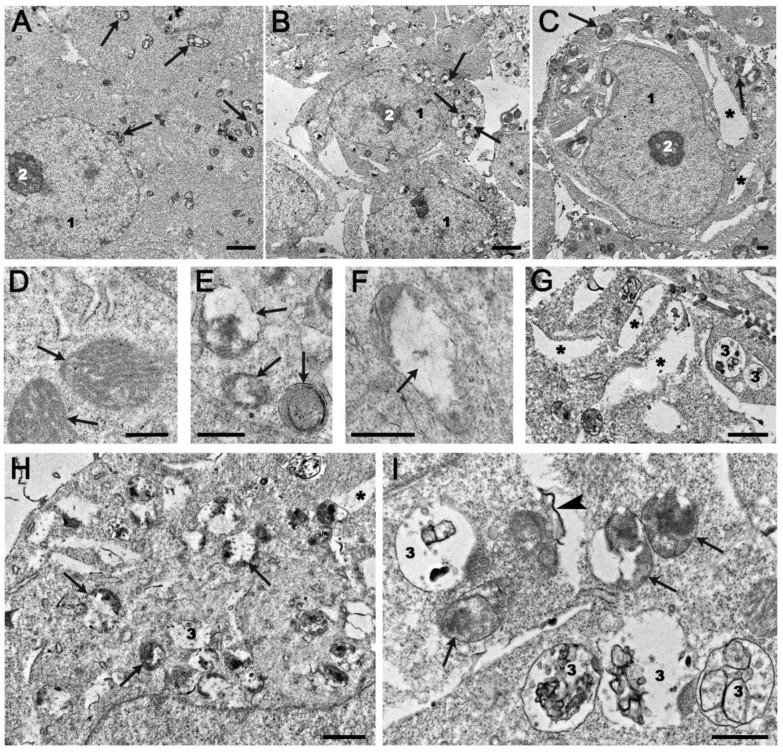
Representative images of HEK293A cells after 24 hours’ incubation: (**B**,**E**,**F**)–topotecan; (**C**,**G**)–OL9-119; (**H**,**I**)–topotecan and OL9-119 mixture. (**A**,**D**)–intact cells. 1—nucleus; 2—nucleolus; 3—late endosomes; the asterisk shows enhanced ER cisternae; the arrows show the mitochondria; the arrowhead shows an electron-dense membrane fragment. TEM, ultrathin sections. The scale bar length corresponds to 2 µm (**A**–**C**), 500 nm (**D**–**F**,**I**) and 1 µm (**G**,**H**).

**Table 1 ijms-26-09291-t001:** Log2FoldChange values of metastasis-related genes with significantly altered expression in TDP1 knockout cells (C6, G6, F7) and in cells treated with OL9-119 (OL9-119 WT, OL9-119 C6, OL9-119 G6, and OL9-119 F7) *.

Gene	C6	G6	F7	OL9-119 WT	OL9-119 C6	OL9-119 G6	OL9-119 F7
*VEGFA*	0.913	1.182	1.017	1.575	0.783	0.789	0.477
*RPSA*	−0.377	−0.356	−0.588	−0.29	-	-	-
*MTSS1*	1.219	1.316	1.915	1.184	-	-	-
*MMP2*	-	−1.232	−1.065	−0.642	-	-	-
*MMP9*	-	1.832	1.957	2.272	-	-	-
*NME4*	−0.733	-	−0.507	−0.517	-	-	-
*TGFB1*	−0.708	−0.599	−1.178	-	-	-	-
*MDM2*	−0.824	−0.454	−0.596	-	-	-	-
*HGF*	4389	3188	3675	-	-	-	-

* *p* (adj) values < 0.05 for all genes.

## Data Availability

Raw data of mRNA sequencing for HEK293A WT cells without treatment and with treatment with DMSO, OL9-119, Topotecan and a combination of OL9-119 + Topotecan: https://www.ncbi.nlm.nih.gov/geo/query/acc.cgi?acc=GSE218871 (1 December 2022); raw data of mRNA sequencing for HEK293A TDP1-KO cell clones C6, G6, and F7 without treatment and treated with OL9-119, Topotecan and their combination: https://www.ncbi.nlm.nih.gov/geo/query/acc.cgi?acc=GSE247361 (20 November 2023). Other raw data are available from the corresponding author upon request.

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
