# Peer review of "Transcriptomic Analysis of TDP1-Knockout HEK293A Cells Treated with the TDP1 Inhibitor (Usnic Acid Derivative)"

_ijms, 2025, doi:10.3390/ijms26199291_

Round 1
Reviewer 1 Report
Comments and Suggestions for Authors
The manuscript «Transcriptomic Analysis of TDP1-Knockout HEK293A Cells treated with TDP1 inhibitor (usnic acid derivative)» by Zakharenko et al. is devoted to the analysis of the effect of inhibitors Top1, Tdp1 and their combination on the transcriptome of HEK293A cells both wild type and TDP1 knockout. The results of the studies showed that treatment with the drugs led to changes in the expression of genes responsible for cell motility, electron transport, mitochondrial function and protein folding. The relevance of this paper is beyond doubt, since despite the achievements in the development of new methods of cancer therapy, traditional chemotherapy remains popular due to its good study and availability, and the possibility of increasing its effectiveness is in high demand. It should be noted that the large amount of data provided is not always informative. For example, Section 2.1.3 provides a fairly extensive list of DEGs without discussing the significance of these changes. Although the article is already quite long, I think it would be a good idea to add several sentences discussing this.
In section 2.5, cell ultrastructural changes indicative of ER stress are observed only under the influence of topotecan, while in section 2.1, a cluster of DEGs under the influence of OL9-119 related to protein folding is described. Is OL9-119 capable of inducing ER stress? This point might be discussed in the text.
Author Response
Comment: The manuscript «Transcriptomic Analysis of TDP1-Knockout HEK293A Cells treated with TDP1 inhibitor (usnic acid derivative)» by Zakharenko et al. is devoted to the analysis of the effect of inhibitors Top1, Tdp1 and their combination on the transcriptome of HEK293A cells both wild type and TDP1 knockout. The results of the studies showed that treatment with the drugs led to changes in the expression of genes responsible for cell motility, electron transport, mitochondrial function and protein folding. The relevance of this paper is beyond doubt, since despite the achievements in the development of new methods of cancer therapy, traditional chemotherapy remains popular due to its good study and availability, and the possibility of increasing its effectiveness is in high demand. It should be noted that the large amount of data provided is not always informative. For example, Section 2.1.3 provides a fairly extensive list of DEGs without discussing the significance of these changes. Although the article is already quite long, I think it would be a good idea to add several sentences discussing this.
Response: The authors thank the Reviewer for the kind words about our manuscript. We fully agree that the presented data set in its current form is not always informative, so we decided to reduce the listing and discussion of individual DEGs and paid more attention discussing the changes in the signaling pathways.
Comment: In section 2.5, cell ultrastructural changes indicative of ER stress are observed only under the influence of topotecan, while in section 2.1, a cluster of DEGs under the influence of OL9-119 related to protein folding is described. Is OL9-119 capable of inducing ER stress? This point might be discussed in the text.
Response: Indeed, despite the large number of ER stress-related DEGs under the influence of OL9-119, we experimentally observed only changes in ER under the influence of topotecan. The apparent contradiction can be explained by the different treatment times of cells for transcriptome (5 hours) and ultramicrographs (24 hours). OL9-119 is an usnic acid derivative. It is known that usnic acid is able to affect processes and cellular structures associated with protein biosynthesis, and we hypothesize that OL9-119 should have a similar effect.
Reviewer 2 Report
Comments and Suggestions for Authors
I appreciate the authors’ efforts in preparing the manuscript titled “Transcriptomic Analysis of TDP1-Knockout HEK293A Cells Treated with a TDP1 Inhibitor (Usnic Acid Derivative)”. This study presents a comprehensive transcriptomic comparison of wild-type (WT) and TDP1-knockout (KO) HEK293A cells following treatment with a TDP1 inhibitor.
However, the manuscript’s structure could be improved by focusing on the major, previously undescribed transcriptomic changes induced by TDP1 inhibitor treatment, rather than discussing each differentially expressed gene (DEG) individually. The current approach dilutes the central message and underemphasizes the study’s main findings. I recommend re-evaluating the most impactful observations from the transcriptomic data, discussing why the inhibitor may influence these cellular pathways, and strengthening the conclusions with experimental validation.
The current results suggest that the TDP1 inhibitor selectively affects genes involved in mitochondrial function. A gene set enrichment analysis (GSEA) could determine whether oxidative phosphorylation–related genes are significantly enriched in the TDP1-treated group, complementing the existing network analysis. Additional validation, such as qPCR or Seahorse assays, would further support the conclusion that TDP1 inhibition disrupts mitochondrial function.
Regarding data presentation, since the manuscript already includes a complete table of DEGs, I suggest removing redundant tables or replacing them with graphical representations to improve readability.
Major Comments:
- Introduction to Results Section:
The opening paragraph of the Results section (“TDP1 knockout samples…”) is not well aligned with the conventional structure. Typically, the Results should begin by restating the biological question and hypothesis, followed by a clear presentation of the key findings. - Biological Replicates:
While I appreciate the use of three independent TDP1-KO clones as biological replicates, the PCA plot suggests notable differences between these clones. The authors should explain the potential causes of this divergence, such as genetic background variation or differences in knockout efficiency. - Rationale for DEG Analyses in 2.1.3:
The strategy for analyzing DEGs across all cell types versus those unique to KO cells needs clarification. Given that OL9-919 is a TDP1 inhibitor, one would expect its effects to parallel those of TDP1 knockout.
-
- If the same DEGs are present across all cell types regardless of TDP1 status, this implies the effects of OL9-919 are largely independent of TDP1 inhibition.
- If DEGs are observed only in KO cells, this could suggest off-target effects.
In either case, these results seem tangential to the manuscript’s stated objective of understanding OL9-919–induced transcriptomic changes, and the focus should be adjusted accordingly.
- Consolidation of Results (Sections 2.1.3–2.4):
The current Results section devotes separate paragraphs to individual genes, which could be more effectively presented as grouped analyses emphasizing pathways and networks. This would make the narrative more concise and impactful.
Minor Comments:
- Figure 1 Title: The PCA plot does not depict “transcriptional patterns” but rather visualizes sample relationships based on principal components. The title should reflect this.
- Interpretation of PCA Plot: The statement “It is evident that the PARP1-KO samples are very different from the others, and that drug treatment has a significantly smaller effect on PARP-KO cells than on TDP1-KO or wild-type cells” cannot be concluded from a PCA plot alone, as statistical significance cannot be inferred from visual clustering. This sentence should be revised.
- Typographical Error in 2.1.3: The word “mutant” should be replaced with “KO” in “... in both wild-type cells and all three mutant clones”.
- Figure 5 Axis Label: The X-axis unit should be µM.
- Compound Name Consistency: OL9-919 appears inconsistently as OL-919 in some places; standardize the nomenclature throughout the manuscript.
- Gene Name Consistency: TDP1 appears inconsistently as Tdp1 in some places; use a consistent format throughout.
Author Response
Comment: I appreciate the authors’ efforts in preparing the manuscript titled “Transcriptomic Analysis of TDP1-Knockout HEK293A Cells Treated with a TDP1 Inhibitor (Usnic Acid Derivative)”. This study presents a comprehensive transcriptomic comparison of wild-type (WT) and TDP1-knockout (KO) HEK293A cells following treatment with a TDP1 inhibitor.
However, the manuscript’s structure could be improved by focusing on the major, previously undescribed transcriptomic changes induced by TDP1 inhibitor treatment, rather than discussing each differentially expressed gene (DEG) individually. The current approach dilutes the central message and underemphasizes the study’s main findings. I recommend re-evaluating the most impactful observations from the transcriptomic data, discussing why the inhibitor may influence these cellular pathways, and strengthening the conclusions with experimental validation.
The current results suggest that the TDP1 inhibitor selectively affects genes involved in mitochondrial function. A gene set enrichment analysis (GSEA) could determine whether oxidative phosphorylation–related genes are significantly enriched in the TDP1-treated group, complementing the existing network analysis. Additional validation, such as qPCR or Seahorse assays, would further support the conclusion that TDP1 inhibition disrupts mitochondrial function.
Regarding data presentation, since the manuscript already includes a complete table of DEGs, I suggest removing redundant tables or replacing them with graphical representations to improve readability.
Response: The authors thank the Reviewer for the useful advices and comments. We hope that taking them into account, the material presented in the article became more structured and better perceived. We improved the manuscript structure trying to focus on transcriptomic changes induced by OL9-119 according to the Reviewer’s comments. We have added new validation experiment on analysis of mitochondrial membrane potential by JC-1 dye (Figure 8), and also gene set enrichment analysis (GSEA) was done (Figure 2D). We also attempted to assess the off-target effect of OL9-119 by comparing the effects of this compound and TDP1 knockout on the transcriptome, as well as by comparing the signaling pathways altered by OL9-119 depending on the TDP1 status of the cells (Figure 5).
Major Comments:
- Introduction to Results Section:
The opening paragraph of the Results section (“TDP1 knockout samples…”) is not well aligned with the conventional structure. Typically, the Results should begin by restating the biological question and hypothesis, followed by a clear presentation of the key findings.
Response: We have added a couple of sentences about the main question of the article at the beginning of the Results section (lines 109 – 112).
- Biological Replicates:
While I appreciate the use of three independent TDP1-KO clones as biological replicates, the PCA plot suggests notable differences between these clones. The authors should explain the potential causes of this divergence, such as genetic background variation or differences in knockout efficiency.
Response: We have added suggestions about the reasons for such diversity among clones (lines 130 – 133).
- Rationale for DEG Analyses in 2.1.3:
The strategy for analyzing DEGs across all cell types versus those unique to KO cells needs clarification. Given that OL9-919 is a TDP1 inhibitor, one would expect its effects to parallel those of TDP1 knockout.
Response: We have significantly expanded the section 2.1.3 (now 2.1.4). We added the comparison of the effect of OL9-119 cells on the HEK293A transcriptome with the effect of TDP1 knockouts.
- If the same DEGs are present across all cell types regardless of TDP1 status, this implies the effects of OL9-919 are largely independent of TDP1 inhibition.
Response: We thank the Reviewer for the valuable comment that highlighted an important aspect of the effects of OL9-119, it may act as a TDP1 inhibitor or as a substance similar to usnic acid which has a large list of biological activities. Thus, the effect of OL9-119 on the transcriptome can be divided into two parts: TDP1-dependent and TDP1-independent DEGs. We have significantly rewritten the results section to emphasize this point.
- If DEGs are observed only in KO cells, this could suggest off-target effects.
In either case, these results seem tangential to the manuscript’s stated objective of understanding OL9-919–induced transcriptomic changes, and the focus should be adjusted accordingly.
Response: We agree with the Reviewer that the off-target effect is not so interesting, especially since there are very few such genes, and we excluded these data.
- Consolidation of Results (Sections 2.1.3–2.4):
The current Results section devotes separate paragraphs to individual genes, which could be more effectively presented as grouped analyses emphasizing pathways and networks. This would make the narrative more concise and impactful.
Response: We fully agree with the Reviewer and have tried to reduce references to individual genes and added information on signaling pathways altered by OL9-119.
Minor Comments:
- Figure 1 Title: The PCA plot does not depict “transcriptional patterns” but rather visualizes sample relationships based on principal components. The title should reflect this.
- Interpretation of PCA Plot: The statement “It is evident that the PARP1-KO samples are very different from the others, and that drug treatment has a significantly smaller effect on PARP-KO cells than on TDP1-KO or wild-type cells” cannot be concluded from a PCA plot alone, as statistical significance cannot be inferred from visual clustering. This sentence should be revised.
- Typographical Error in 2.1.3: The word “mutant” should be replaced with “KO” in “... in both wild-type cells and all three mutant clones”.
- Figure 5 Axis Label: The X-axis unit should be µM.
- Compound Name Consistency: OL9-919 appears inconsistently as OL-919 in some places; standardize the nomenclature throughout the manuscript.
- Gene Name Consistency: TDP1 appears inconsistently as Tdp1 in some places; use a consistent format throughout.
Response: We took into account all the Reviewer's Minor comments and made the necessary changes.
Round 2
Reviewer 2 Report
Comments and Suggestions for Authors
The authors have addressed all my comments. I recommend publication as is.